# Outcomes and Timing of Bedside Percutaneous Tracheostomy of COVID-19 Patients over a Year in the Intensive Care Unit

**DOI:** 10.3390/jcm10153335

**Published:** 2021-07-28

**Authors:** Nardi Tetaj, Micaela Maritti, Giulia Stazi, Maria Cristina Marini, Daniele Centanni, Gabriele Garotto, Ilaria Caravella, Cristina Dantimi, Matteo Fusetti, Carmen Santagata, Manuela Macchione, Giada De Angelis, Filippo Giansante, Donatella Busso, Rachele Di Lorenzo, Silvana Scarcia, Alessandro Carucci, Ricardo Cabas, Ilaria Gaviano, Nicola Petrosillo, Andrea Antinori, Fabrizio Palmieri, Gianpiero D’Offizi, Stefania Ianniello, Paolo Campioni, Francesco Pugliese, Francesco Vaia, Emanuele Nicastri, Giuseppe Ippolito, Luisa Marchioni

**Affiliations:** 1UOC Resuscitation, Intensive and Sub-Intensive Care, National Institute for Infectious Diseases IRCCS, Lazzaro Spallanzani, 00149 Rome, Italy; micaela.maritti@inmi.it (M.M.); giuliavaleria.stazi@inmi.it (G.S.); mariacristina.marini@inmi.it (M.C.M.); gabriele.garotto@inmi.it (G.G.); ilaria.caravella@inmi.it (I.C.); cristina.dantimi@inmi.it (C.D.); matteo.fusetti@inmi.it (M.F.); carmen.santagata@inmi.it (C.S.); manuela.macchione@inmi.it (M.M.); giada.deangelis@inmi.it (G.D.A.); filippo.giansante@inmi.it (F.G.); donatella.busso@inmi.it (D.B.); rachele.dilorenzo@inmi.it (R.D.L.); silvana.scarciadaprano@inmi.it (S.S.); alessandro.carucci@inmi.it (A.C.); ricardo.cabas@inmi.it (R.C.); Ilaria.gaviano@inmi.it (I.G.); luisa.marchioni@inmi.it (L.M.); 2Clinical and Research Department of Infectious Diseases, National Institute for Infectious Diseases IRCCS Lazzaro Spallanzani, 00149 Rome, Italy; daniele.centanni@inmi.it (D.C.); nicola.petrosillo@inmi.it (N.P.); andrea.antinori@inmi.it (A.A.); Fabrizio.palmieri@inmi.it (F.P.); gianpiero.doffizi@inmi.it (G.D.); Emanuele.nicastri@inmi.it (E.N.); 3Department of Radiology and Diagnostic Imaging, National Institute for Infectious Diseases IRCCS Lazzaro Spallanzani, 00149 Rome, Italy; stefania.ianniello@inmi.it (S.I.); paolo.campioni@inmi.it (P.C.); 4Department of Anesthesia and Critical Care Medicine, Sapienza University of Rome, 00161 Rome, Italy; f.pugliese@uniroma1.it; 5Health Direction, National Institute for Infectious Diseases IRCCS Lazzaro Spallanzani, 00149 Rome, Italy; francesco.vaia@inmi.it; 6Scientific Direction, National Institute for Infectious Diseases IRCCS Lazzaro Spallanzani, 00149 Rome, Italy; giuseppe.ippolito@inmi.it

**Keywords:** intensive care unit, percutaneous tracheostomy, COVID-19, ICU stay, early tracheostomy, late tracheostomy, ICU length of stay, healthcare workers, mechanical ventilation

## Abstract

Background: The benefits and timing of percutaneous dilatational tracheostomy (PDT) in Intensive Care Unit (ICU) COVID-19 patients are still controversial. PDT is considered a high-risk procedure for the transmission of SARS-CoV-2 to healthcare workers (HCWs). The present study analyzed the optimal timing of PDT, the clinical outcomes of patients undergoing PDT, and the safety of HCWs performing PDT. Methods: Of the 133 COVID-19 patients who underwent PDT in our ICU from 1 April 2020 to 31 March 2021, 13 patients were excluded, and 120 patients were enrolled. A trained medical team was dedicated to the PDT procedure. Demographic, clinical history, and outcome data were collected. Patients who underwent PDT were stratified into two groups: an early group (PDT ≤ 12 days after orotracheal intubation (OTI) and a late group (>12 days after OTI). An HCW surveillance program was also performed. Results: The early group included 61 patients and the late group included 59 patients. The early group patients had a shorter ICU length of stay and fewer days of mechanical ventilation than the late group (*p* < 0.001). On day 7 after tracheostomy, early group patients required fewer intravenous anesthetic drugs and experienced an improvement of the ventilation parameters PaO_2_/FiO_2_ ratio, PEEP, and FiO_2_ (*p* < 0.001). No difference in the case fatality ratio between the two groups was observed. No SARS-CoV-2 infections were reported in the HCWs performing the PDTs. Conclusions: PDT was safe and effective for COVID-19 patients since it improved respiratory support parameters, reduced ICU length of stay and duration of mechanical ventilation, and optimized the weaning process. The procedure was safe for all HCWs involved in the dedicated medical team. The development of standardized early PDT protocols should be implemented, and PDT could be considered a first-line approach in ICU COVID-19 patients requiring prolonged mechanical ventilation.

## 1. Introduction

In December 2019, a novel viral agent named Severe Acute Respiratory Syndrome Coronavirus-2 (SARS-CoV-2) was identified as the etiologic agent of the Coronavirus Disease 2019 (COVID-19) outbreak occurring in Wuhan, China [1].

SARS-CoV-2 is transmitted from person to person via droplets, contact, and aerosolized particles [2].

Patients with progressive disease should be monitored closely for worsening respiratory status; they typically require supplemental oxygen to maintain an oxygen saturation (SpO_2_) of ≥94% [3].

Orotracheal intubation (OTI) and mechanical ventilation can be used in patients with more severe cases affected by acute respiratory distress syndrome (ARDS) in the Intensive Care Unit (ICU) [3].

In the ICU, prolonged mechanical ventilation and weaning failure from ventilator support are the most frequent criteria for the indication of tracheostomy [4]. 

Tracheostomy has many beneficial effects, such as improving pulmonary mechanics; reducing laryngeal or tracheal nociceptive stimuli; facilitating the weaning process; and decreasing the requirement for sedatives, neuromuscular blocker agents, analgesics, and inotropic therapy. It also reduces dead space and airway resistance, helps maintain easier oral hygiene, promotes oral nutrition, and improves communication [5,6].

Decannulation in tracheostomized patients is the final step towards liberation from mechanical ventilation and should be attempted as soon as possible [7].

The tracheostomy procedure is associated with an increased risk of aerosol generation, and it is recommended to be performed in an airborne isolation room with all healthcare workers (HCWs) involved wearing full adequate personal protective equipment (PPE) [8].

Recommendations on both safety and timing protocols for tracheostomy in COVID-19 patients have been published, but there are still controversial issues [9].

The aim of this study was to investigate the following outcomes in COVID-19 ICU patients: −The benefits of percutaneous dilatational tracheostomy (PDT) on respiratory functions and the weaning process−The timing of tracheostomy and its association with ICU length of stay and mechanical ventilation−Risk of SARS-CoV-2 infection in HCWs during the PDT procedure

## 2. Materials and Methods

### 2.1. Study Design and Participants

This was a retrospective observational study conducted at the National Institute for Infectious Disease (INMI) “Lazzaro Spallanzani” in Rome, Italy, which is an over 200-bed hospital for infectious disease with a 55-bed ICU. In this study, we included adult patients with respiratory failure due to COVID-19 hospitalized in our Intensive Care Unit (ICU) from 1 April 2020 to 31 March 2021. 

During this period, 2480 patients with virologically confirmed COVID-19 by nasal pharyngeal swab for reverse transcriptase polymerase chain reaction (rtPCR) assay were hospitalized to our hospital, of which 451 were admitted to the ICU and 342 underwent orotracheal intubation (OTI).

We performed 133 percutaneous dilatation tracheostomies (PDTs) at bedside in the intubated patients’ ICU rooms using the Frova PercuTwist technique with rotational dilatation of the tracheal stoma through the use of hydrophilic screws [10] or Ciaglia Blue Rhino using of a single-beveled curved hydrophilic dilator [11]. Both techniques were conducted under video-assisted fibro-bronchoscopy. 

The inclusion criteria for tracheostomy were acute respiratory failure and the need for prolonged mechanical ventilation.

The exclusion criteria for tracheostomy were infection at the site of the tracheostomy, uncontrolled coagulopathy, altered neck anatomy, marked obesity, and multiorgan failure (MOF) [12].

Tracheostomies were performed by a medical team of highly trained HCWs including 16 anesthesiologists and 8 nurses. Each individual PDT procedure was performed by a single team unit including two anesthesiologists and two nurses as follows: (1) the first anesthesiologist performed the fibrobronchoscopy; (2) the second anesthesiologist performed the tracheostomy; (3) two nurses managed the ventilator and endotracheal tube and administered medications for sedation and curarization.

The performed PDT procedure can be briefly described in the following phases. The bronchoscope was inserted through the mount catheter without suspending mechanical ventilation in order to avoid hypoxia in the patient, and the fraction of inspired oxygen (FiO_2_) was set at 100% during the procedure. Once the space between the 1st and 2nd or the 2nd and 3rd tracheal rings was identified by transillumination under direct vision through video-bronchoscopy, the bronchoscopist operator retracted the endotracheal tube (ETT) up close to the first tracheal ring, maintaining a good view of the tracheal lumen. The second operator in the sterile field used the aseptic technique to proceed with spy needle insertion into the tracheal space identified, and through the Seldinger wire technique performed the dilation of the stoma and the insertion of the tracheal cannula without stopping mechanical ventilation. Once inserted, the tracheal cannula was cuffed and connected to the mechanical ventilator while the endotracheal tube was removed; this switch lasted a few seconds to avoid patient hypoxia from apnea.

According to the international guidelines and institutional policies regarding HCW protection [2], the team wore full personal protection equipment (PPE) during the procedures, including N95 masks/FFP3 masks, surgical masks, headgear, hoods, eye protection, powered air-purifying respirators (PAPRs), long-sleeved water-repellent shirts, and double gloves, upon entering an airborne isolation room. The room was a dedicated negative pressure room with at least 6 air changes per hour.

Patients who underwent bedside PDT in the ICU were stratified into two groups: (1)The early group, which included patients who underwent PDT within the first 12 days of orotracheal intubation (OTI)(2)The late group, which included patients in whom the procedure was performed more than 12 days after OTI.

### 2.2. Data Collection

The data collected included age, gender, BMI (body mass index), SOFA (sequential organ failure assessment) score, APACHE II score (acute physiology and chronic health evaluation) at ICU admission, days of hospitalization pre-ICU admission, previous hospitalization within the last 6 months, previous surgical procedures in last month, and comorbidities (arterial hypertension, other cardiac diseases, diabetes, kidney disease (stage 3–5 of CKD), moderate to severe liver disease, chronic obstructive pulmonary disease or bronchial asthma, solid neoplasia or hematological malignancy within the last 5 years, chronic neurological disorders, autoimmune diseases, obesity (defined as BMI > 30 kg/m^2^), and other diseases). 

During the ICU stay, we also recorded the date of the endotracheal intubation, tracheostomy, weaning and decannulation, and ICU outcome (discharge or exitus).

Ventilation parameter data were recorded on the day of orotracheal intubation (OTI), percutaneous tracheostomy (PDT), and seven days after the PDT; these data included the respiratory exchange ratio (PaO_2_/FiO_2_ ratio), the fraction of inspired oxygen (FiO_2_), and positive end-expiratory pressure (PEEP).

Data collected on medications administered by intravenous continuous infusion (c.i.v.) included doses of sedatives, opioids as analgesics, and inotropic agents.

The complications of tracheostomy were assessed during the procedure and throughout the hospital stay.

All tracheostomized patients received pulmonary and physical rehabilitation starting from the beginning of weaning from mechanical ventilation.

All involved HCWs periodically underwent internal training courses to ensure effective infection prevention and control (IPC) measures. This identified logistical and technical challenges in the intensive care unit considering the possibility of airborne transmission, especially through aerosol-generating procedures such as airway suctioning, nebulizer treatment, noninvasive ventilation (NIV), bronchoscopy, intubation, and tracheotomy. 

Active surveillance and early identification of suspected cases of COVID-19 among HCWs were established.

Serological SARS-CoV-2 tests and nasopharyngeal swab tests for the molecular detection of SARS-CoV-2 (rtPCR) were performed for the surveillance of all HCWs 14 days after performing a PDT and thereafter at least once every three weeks, unless they had symptoms or had been in direct contact with confirmed SARS-CoV-2 patients.

### 2.3. Statistical Analysis

Quantitative variables are expressed as medians (interquartile range, IQR), means (±standard deviation, SD), and 95% confidence intervals (95% CI). The means of quantitative variables were compared with Student’s *t*-tests and used in repeated measures ANOVA over the timeline. Nominal data are expressed as N (percentages, %).

Adjusted estimates of ICU length of stay and total days on mechanical ventilation were obtained for both the early and late PDT groups using multivariable linear regression models carried out by the STATA 15 statistical package (Stata Corp LP, College Station, TX, USA).

Statistical confounding factors selection was performed through backward elimination, removing from the model all nonsignificant confounders (*p*-value > 0.05). 

In the end, two final regression models were presented: the first used ICU length of stay as a dependent variable and early/late group, neurological disorders, hypertension, and age categories (<60, 60–70, >70 years) as independent variables; the second model involved the total days on mechanical ventilation as the dependent variable and early/late group, hypertension, diabetes, and age categories (<60, 60–70, >70 years) as independent variables. Eventually, independent *t*-tests between the early and late PDT groups for both ICU length of stay and the total days on mechanical ventilation estimates were performed.

## 3. Results

### 3.1. Outcome 7 Days after PDT in COVID-19 ICU Patients

During the study period, 2480 COVID-19 patients were hospitalized at our center; 451 (18.2%) of them were admitted to our ICU, 342 (76%) patients underwent OTI, and of these, 133 patients (38.9%) underwent bedside PDT.

We excluded from the study 13 patients: 8 patients were transferred to other hospitals, 4 patients were transferred to an extracorporeal membrane oxygenation center, 3 patients were transferred to a coronary care unit, and 1 patient was transferred to a surgical care unit; five patients were still admitted at the end of the study. 

Finally, we studied 120 COVID-19 ICU patients (Figure 1).

The median age was 68.5 years (IQR, 61–75), 66.7% were male, and the median body mass index (BMI) was 27.8 (IQR, 25–31). The comorbidities with the highest prevalence were arterial hypertension (66.7%), obesity (45%), other heart diseases (24%), and diabetes (23.3%) (Table 1).

The mean time of pre-ICU hospitalization of the cohort was 6 days (range: 4.3–7.5 days, 95% CI); PDT was performed a median of 12 days and a mean of 14 days (± 8.9, SD) after OTI (Figure 2). We decided on day 12 after OTI as the point of demarcation between early and late tracheostomies using the median value of PDT in all our patients. Additionally, data from several studies and reviews of early vs. late tracheostomy recommend a range of 10–14 days [9].

Ten patients died within seven days of tracheostomy. We excluded them from the study analysis on administered drug dosage and ventilation parameters seven days after PDT (Table 2).

A significant temporal association between PDT insertion and the use of sedative, analgesic, and inotropic drugs was observed; propofol, remifentanil, and norepinephrine were all significantly reduced on day 7 after PDT (*p* < 0.001 for all parameters, see Table 2 and Figure 3).

Furthermore, a significant association between PDT insertion and improvement of common ventilation parameters (PaO_2_/FiO_2_ ratio, PEEP, and FiO_2_) on day 7 after PDT was observed (*p* < 0.001 for all parameters, see Table 2 and Figure 4).

Table 3 summarizes the clinical progress of COVID-19 patients during their ICU stays. The mean time on mechanical ventilation was 34.5 days (32–37 days, 95% CI). The mean length of stay in the ICU was 42 days (39–46, 95% CI). The mean time on tracheostomy, from its insertion to decannulation, was 26.7 days (24–29, 95% CI).

The mean time to spontaneous breathing from the start of weaning to decannulation was 6.1 days (5–7, 95% CI).

The case fatality rate (CFR) during the ICU stay was 35% (42 patients), due to severe respiratory failure (13 patients, 31%), septic shock (14 patients, 33.3%) [13], multiple organ failure (MOF) (10 patients, 23.8%), massive pulmonary embolism (4 patients, 9.5%), and cerebral venous thrombosis (1 patient, 2.3%). 

### 3.2. PDT Complications

The PDT procedure lasted for 10 to 20 min. No early complications during the insertion of the tracheostomy were observed. Periprocedural and postprocedural complications were assessed during the ICU stay until hospital discharge. On day 7 after PDT, subcutaneous emphysema limited to the neck was present in three patients (2.7%). All patients were treated with therapeutic dose anticoagulants, and 23 patients (21%) presented spontaneously resolved minor bleeding at the stoma site (of them, nine were treated in association with antiplatelet drugs). All patients underwent motor and respiratory physiotherapy during their stay in the ICU after weaning and continued it after discharge.

On day 60 after PDT, three patients were still under tracheostomy but with spontaneous breathing followed by accomplishment of decannulation. 

Twenty-three patients were discharged to the acute care unit with the tracheostomy cannula still in place, and subsequently, all were successfully decannulated. At 90 days after PDT, all surviving patients (78 patients, 71%) were decannulated and no complications were observed. 

After decannulation, 11 patients (14%) had difficulty swallowing and underwent logopedic therapy with full recovery by 1 month after decannulation.

### 3.3. Early versus Late PDT

Sixty-one patients underwent early PDT within 12 days of orotracheal intubation (OTI), with a mean of 7.6 days (7–8.3 days, 95% CI) (Table 3 and Figure 5).

Fifty-nine patients underwent late PDT (>12 days after OTI), with a mean of 20 days (18–22.5 days, 95% CI). 

Regarding the ICU length of stay (LOS), the mean ICU LOS of 37.7 days (95% CI, 36–39 days) was significantly lower than the mean LOS in the late group, which was 47.7 days (95% CI, 46–49 days), (*p* < 0.001).

As expected, similar data were obtained for the time on mechanical ventilation (MV) between the two groups: patients in the early group were exposed to MV for a mean of 30 days (95% CI, 29–31 days), significantly shorter than the patients in late group, who underwent MV with a mean of 39 days (95% CI, 38–40 days) (*p* < 0.001).

The boxplots of ICU LOS and total days on MV are reported in Figure 5.

There was no statistically significant difference in the mortality between the early (23 patients, 37.7%) and late group (19 patients, 32.2%) (*p* = 0.165).

### 3.4. SARS-CoV-2 Transmission in HCWs Who Performed PDT 

No team members performing the PDT procedure developed symptoms related to SARS-CoV-2 infection, and all serial serology tests to detect IgG and IgM antibodies against SARS-CoV-2 were persistently negative. Additionally, all nasopharyngeal swabs for the molecular detection of SARS-CoV-2 were negative on day 14 after the PDT procedure, periodically every three weeks, and in all cases of suspected symptoms or close contact with SARS-CoV-2 positive cases. In January 2021, all ICU HCWs were vaccinated early during the vaccination campaign promoted by the Italian Ministry of Health.

## 4. Discussion

The PDT procedure is safe for both healthcare professionals and COVID-19 patients. Early PDT procedures had a beneficial impact on COVID ICU patients in terms of ICU LOS and time on MV in order to both improve pulmonary performance and reduce the weaning process.

During mechanical ventilation with the endotracheal tube (ETT) in the ICU, all patients were sedated with drugs in continuous intravenous infusion using propofol and remifentanil or, in case of an unstable hemodynamic situation, using norepinephrine. In 10 cases, continuously infused midazolam was used as an additional sedative agent for less than 48 h. Likewise, curarization was used for less than 48 h. On day 7 after the PDT procedure, a significantly decreased dose of continuous intravenous sedation and inotropic therapy was observed (Figure 1 and Table 2). 

No patients required surgical tracheostomy during the study period. Percutaneous tracheotomy should be preferred over the surgical technique in intensive care patients, as this recommendation has a high level of evidence (GRADE 1+/strong agreement). In addition, PDT is associated with a shorter operative time and a decreased incidence of stoma infection, inflammation, and postprocedural major bleeding [14].

One additional clinical benefit of PDT compared with ETT is the lack of pharyngeal and laryngeal stimuli due to the ETT and a reduction in tracheal stimuli. This allows a gradual reduction in the continuously infused sedative agent and the dosages of the inotropic agent used to counteract arterial hypotension. Furthermore, there was a statistically significant improvement in ventilation support in terms of the PaO_2_/FiO_2_ ratio, FiO_2_, and PEEP, all relevant parameters to increase lung performance and achieve earlier recovery in ICU COVID-19 patients.

In a few studies, it has been shown that propofol, remifentanil, midazolam, and curare could be responsible for the patient’s immunodepression and, at the same time, for a prolonged release of cytokines, micro-aspiration, abnormal peristalsis, microcirculatory effects, and predisposition to infections [15,16]. In addition, the acceleration of the weaning process appears to decrease tracheobronchial colonization by pathogens and the incidence of ventilator-associated pneumonia (VAP) [17]. After weaning in ICU, motor and respiratory rehabilitation through the use of incentive spirometry also had an important impact on accelerating the patient’s recovery. Finally, decannulation of ICU COVID-19 patients was performed successfully and without complications after 30 days.

An early tracheostomy accelerates the weaning process, reduces the predisposition to VAP and systemic infections, accelerates the patient’s awakening and motility, and facilitates nursing care.

The present study showed that early group patients (PDT ≤ 12 days after OTI) had a shorter ICU LOS and were more prone to be ventilator-free. Nevertheless, early PDT did not significantly change the CFR during ICU stay.

Instruction and education of the healthcare professionals with a focus on infection control and prevention measures—particularly on aerosol-generating procedures—are key to ensure the safety of healthcare professionals and protect patients.

Although healthcare professionals who participated in PDT and OTI procedures had the highest increased risk of being infected by SARS-CoV-2, tracheostomy in COVID-19 ICU patients within a comprehensive and integrated training program appears to be a safe procedure. Over the entire study period, no healthcare professionals tested positive for SARS-CoV-2 by nasopharyngeal swabs for molecular detection or serology tests for IgG and IgM antibodies, even before COVID-19 vaccination.

Before drawing conclusions, several limitations are considered. First, this is a longitudinal observational study designed in a single healthcare facility, not a blinded study; this is because the decision about the timing of the tracheostomy was judged on the basis of the opinion of the intensive care physician and the clinical status of the patient. Second, controlling for confounders in any observational study may be incomplete despite all efforts, and further research on larger groups of patients is required. However, this study also has some strengths, including a large sample size and the full representativeness of real-life management of ICU COVID-19 patients with severe respiratory failure.

## 5. Conclusions

In summary, the development of early PDT protocols in COVID-19 ICU cases could be beneficial in reducing the length of ICU stay and the duration of mechanical ventilation. Early PDT should be considered as a first-line approach in COVID-19 ICU patients with severe respiratory failure. Early PDT is a fast and safe procedure performed at the patient’s bedside and avoids unnecessary transport of the patient to the operating room for open surgical tracheostomy. The decannulation of the patients was successfully performed with no major complications. Finally, the nursing management of COVID-19 patients with tracheostomy was more comfortable. 

## Figures and Tables

**Figure 1 jcm-10-03335-f001:**
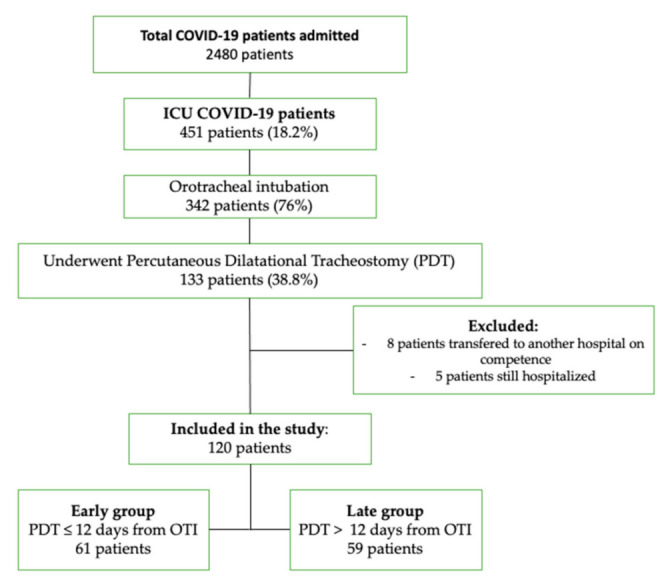
Flowchart of study selection. Abbreviations: ICU, intensive care unit; PDT, percutaneous dilatational tracheostomy; OTI, orotracheal intubation.

**Figure 2 jcm-10-03335-f002:**
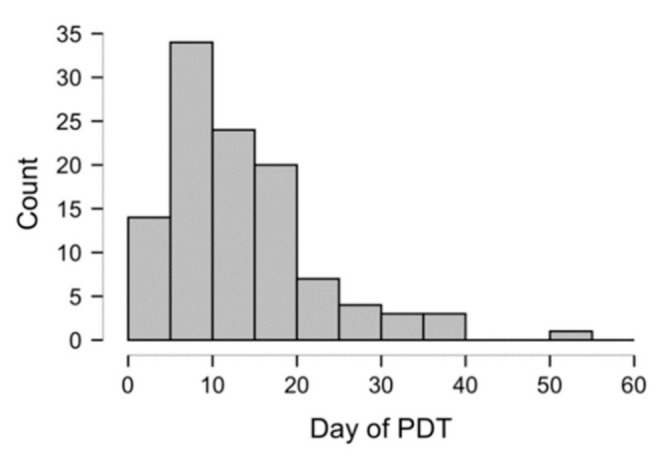
Frequency distribution of percutaneous dilatational tracheostomy (PDT) from orotracheal intubation (OTI).

**Figure 3 jcm-10-03335-f003:**
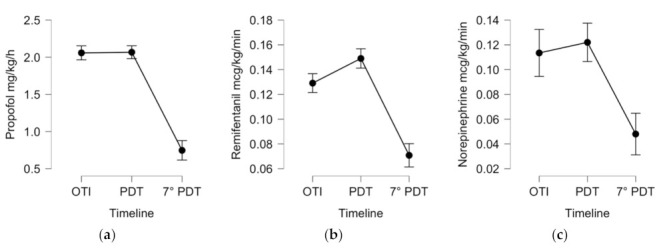
Descriptive plots, means, and 95% confidence intervals of 110 ICU COVID-19 patients who underwent PDT using repeated measures ANOVA for the following variables: Propofol (**a**), Remifentanil (**b**), and Norepinephrine (**c**) administered by intravenous continuous infusion during the ICU stay. Legend: OTI, day of orotracheal intubation; PDT, day of percutaneous tracheostomy; 7° PDT, day 7 after tracheostomy.

**Figure 4 jcm-10-03335-f004:**
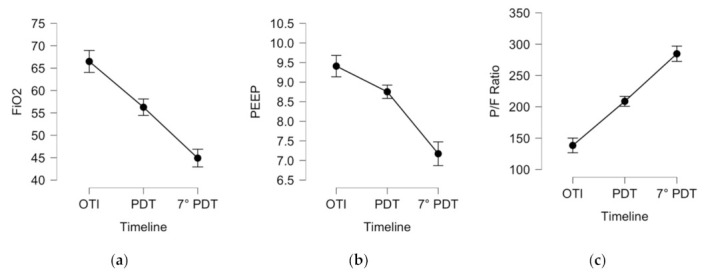
Descriptive plots, means, and 95% confidence intervals of 110 ICU COVID-19 patients underwent PDT, for the following ventilation variables: FiO_2_ (%), fraction of inspired oxygen (**a**); PEEP (cmH_2_O), positive end-expiratory pressure (**b**); P/F ratio (mmHg), respiratory exchange ratio (**c**). Legend: OTI, the day of orotracheal intubation; PDT, the day of percutaneous tracheostomy; 7° PDT, day 7 after tracheostomy.

**Figure 5 jcm-10-03335-f005:**
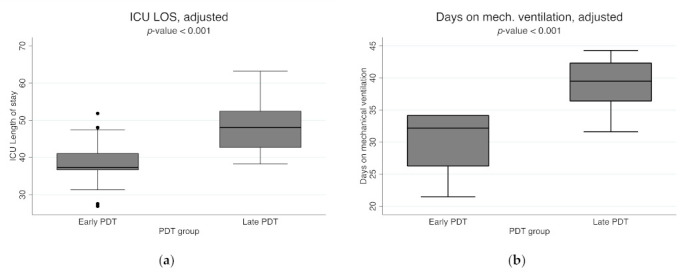
Boxplot of ICU length of stay (**a**) and days of mechanical ventilation (**b**) estimates, adjusted for confounding factors between the two groups (early group and late group). Legend: PDT, percutaneous dilatational tracheostomy; ICU LOS, Intensive Care Unit length of stay; early group, the group of patients who underwent PDT within the first 12 days of orotracheal intubation; late group, the group of patients who underwent PDT more than 12 days after orotracheal intubation.

**Table 1 jcm-10-03335-t001:** The baseline characteristics of ICU COVID-19 patients who underwent PDT.

	Total (*n* = 120)	Early Group (*n* = 61)	Late Group (*n* = 59)	*p*-Value ^a^
Age, median (IQR)	68.5 (61–75)	70 (64–77)	65 (69–73)	0.056
Male, *n* (%)	80 (66.7%)	42 (68.9%)	38 (64.4%)	0.606
Female, *n* (%)	40 (33.3%)	19 (31.1%)	21 (35.5%)	
BMI, kg/m^2^, median (IQR)	27.8 (25–31)	27.6 (25–31)	28.4 (26–31)	0.357
SOFA score, median (IQR)	5 (3–7)	5 (4–7)	5 (3–8)	0.817
APACHE II score, median (IQR)	12.5 (10–17)	13 (10–17)	12 (9–17)	0.646
Comorbidities, *n* (%)				
Arterial hypertension	80 (66.7%)	45 (73.7%)	35 (59.3%)	0.093
Other cardiopathies	24 (20.0%)	15 (24.6%)	9 (15.2%)	0.201
Diabetes	28 (23.3%)	15 (24.6%)	13 (22.0%)	0.741
Obesity	54 (45.0%)	25 (41%)	29 (49.1%)	0.369
Kidney disease (stage 3–5 of CKD)	8 (6.7%)	5 (8.2%)	3 (5.1%)	0.494
Moderate to severe chronic liver disease	1 (0.8%)	1 (1.6%)	0 (0.0%)	0.323
COPD/Bronchial asthma	21 (17.5%)	12 (19.7%)	9 (15.2%)	0.524
Previous neoplasia (solid neoplasia or hematological malignancy in the last 5 years)	14 (11.7%)	7 (11.4%)	7 (11.8%)	0.621
Previous surgery in last month	4 (3.3%)	2 (3.2%)	2 (3.4%)	0.973
Previous hospitalization last six months	8 (6.7%)	3 (4.9%)	5 (8.5%)	0.435
Chronic neurological disorders	20 (16.7%)	10 (16.4%)	10 (16.9%)	0.935
Autoimmune diseases	15 (12.5%)	6 (9.8%)	9 (15.2%)	0.370
Other chronical diseases	32 (26.6%)	23 (37.7%)	9 (15.2%)	0.005

Abbreviations: ICU, intensive care unit; BMI, body mass index; SOFA score, sequential organ failure assessment; APACHE II score, acute physiologic and chronic health evaluation at ICU admission; CKD, stages of chronic kidney disease; COPD, chronic obstructive pulmonary disease; IQR, interquartile range; ^a^ Chi square test was performed between the early and late groups.

**Table 2 jcm-10-03335-t002:** Characteristics of laboratory, ventilation, and infusion therapy parameters.

Total ^1^ (*n* = 110), Mean ± SD	Day of OTI in ICU	Day of PDT	Day 7 after PDT	*p*-Value ^a^
**Sedative, analgesic, and inotropic therapy by intravenous continuous infusion**				
**Propofol, mg/kg/h**	2.0 (±0.39)	2.01 (±0.53)	0.7 (±0.87)	*p* < 0.001
**Remifentanil, mcg/kg/min**	0.13 (±0.03)	0.15 (±0.05)	0.07 (±0.07)	*p* < 0.001
**Norepinephrine, mcg/kg/min**	0.11 (±0.12)	0.12 (±0.13)	0.05 (±0.09)	*p* < 0.001
**Ventilation parameters and respiratory exchange ratio**				
**P/F Ratio, mmHg**	138 (±43)	208 (±64)	243 (±95)	*p* < 0.001
**FiO_2_, %**	66.5 (±15)	56 (±14)	45 (±14)	*p* < 0.001
**PEEP, cmH_2_O**	9.4 (±1.9)	8.7 (±1.5)	7.1 (±1.6)	*p* < 0.001

Abbreviations: ICU, intensive care unit; PDT, percutaneous dilatational tracheostomy; FiO_2_, fraction of inspired oxygen; PEEP, positive end-expiration pressure; P/F Ratio, respiratory exchange ratio. ^a^ Paired samples *t*-test of parameters of ICU COVID-19 patients between the day of the PDT and seven days after the PDT; ^1^ PDT cohort without the 10 patients who died within 7 days after PDT.

**Table 3 jcm-10-03335-t003:** The clinical characteristics during patient ICU stay.

	Total ^1^ (*n* = 120) Mean ± SD	Early Group (*n* = 61), Mean ± SD	Late Group (*n* = 59), Mean ± SD	*p*-Value ^a^	*p*-Value ^b^ Adjusted
Pre-ICU hospitalization, days	5.9 (±8.0)	5.6 (±6.5)	6.2 (±9.4)	0.819	
Day of PDT, days from OTI to PDT	13.9 (±8.7)	7.6 (±2.6)	20.4 (±8.1)	<0.001	
Mechanical ventilation, days	34.5 (±15.8)				
Mean		30.3 (±14.1)	38.9 (±16.3)	<0.001	
Adjusted mean **^b^**		30.3 (±4.2) **^b^**	38.9 (±4.3) **^b^**		<0.001
Spontaneous breathing from weaning to decannulation, days	6.1 (±5.5)	5.2 (±4.9)	6.9 (±6.1)	0.085	
Days on tracheostomy, from PDT to decannulation	26.7 (±16.3)	27.3 (±16.2)	25.9 (±16.7)	0.650	
ICU length of stay, days	42.3 (±18.9)				
Mean		36.8 (±17.1)	48.4 (±19.4)	<0.001	
Adjusted mean **^b^**		37.7 (±6.6) **^b^**	47.7 (±6.5) **^b^**		<0.001
Outcome					
ICU discharged, patients (%)	78 (65%)	38 (62.3%)	40 (67.8%)		
ICU mortality, patients (%)	42 (35%)	23 (37.7%)	19 (32.2%)	0.165	

Abbreviations: ICU, intensive care unit; OTI, orotracheal intubation; PDT, percutaneous dilatational tracheostomy. ^1^ Total PDT cohort including the 10 patients who died within 7 days of PDT; ^a^ independent samples *t*-test comparing the means between the early and late groups; **^b^** adjusted for confounding factors between the two groups using multivariate linear regression models.

## Data Availability

The data presented in this study are available on request from the corresponding author. The data are not publicly available because of patient privacy and data protection regulations.

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
