# Peer review of "Outcomes and Timing of Bedside Percutaneous Tracheostomy of COVID-19 Patients over a Year in the Intensive Care Unit"

_jcm, 2021, doi:10.3390/jcm10153335_

Round 1

Reviewer 1 Report

  1. It would be better to refine the English structure or expression.
  2. The authors presented three objectives of this study in the introduction, but a clear interpretation or explanation of optimal timing is lacking. Explain why the authors divided the criteria for early and late by 12 days. 
  3. In the flowchart of study selection, 10 people who died within 7 days after PDT were excluded. However, since table 3 was included the analysis of ICU mortality and the day of PDT in early group was 7.7±2.6 days, the analysis of table 3 should include these 10 patients. Please mention the reason for excluding these 10 patients in the discussion. 

Author Response

Dear Reviewer, we thank you for your valuable suggestions.

We agreed on day 12 from OTI as point of demarcation between early and late tracheostomy, by using the median value of PDT in all our patients. It summarizes data from several studies and reviews of early vs late tracheostomy recommending a range of 10-14 days (added on rows 212-215 of the manuscript).

We took your advice by including the 10 patients who died within day 7 from the PDT, and processed the data collected in view of 120 patients, Table 1 and Table 3.

In the analysis of drugs and ventilatory parameters we had to exclude these 10 patients because the analysis is done considering the seventh day from PDT, Table 2. 

In the attached manuscript, we've highlighted the changes in yellow.

Please feel free to contact me if you have any question.

Kind regards,

Nardi Tetaj

Reviewer 2 Report

I congratulate with the author for the interesting and up to date paper and the good number of patients enrolled. I have some questions to improve the qulity of the paper.

The author should briefly discuss the reason for early versus late intubation time and the advantages of PDT versus surgical tracheostomy (this may help readers). They do not report any significative bleeding problems, so they should clarify the anticoagulation method used (profilactic versus therapeutic or associated with antiplatelet drugs).

The author should display the total number of intubated patients just to be able to asses how many patients were weaned without tracheostomy both in the early group and in the late one. Has tracheostomy been performed in the most of tubed patients?. The difference in FiO2, PEEP and P/F ratio should be compared between early versus late group just to evidence the impact of prolonged intubation on lung mechanics. The author should show if there are difference in lung sovrainfections (with the microbiological culture if any), just to asses if early tracheostomy reduced incidence of VAP.

Despite early tracheostomy group had an early ICU discharge, late tracheostomy had a positive trend on survival benefit. The author should try to give some explanation (does early discharge impose a overload work on ward ? a longer ICU stay offer a safe monitoring...The paper of Prof Terragni et al may help JAMA 2010 Apr 21;303(15):1483-9)

I think that the paper will help readers in future decision but should be upgradated

ty

Author Response

Dear Reviewer, we thank you for your valuable suggestions. 

We took your advice and made some changes on the manuscript. Below we will try to answer your questions as best we can:

  • No patient required surgical tracheostomy during the study period. Percutaneous tracheotomy should be preferred to surgical technique in intensive care patients, the recommendation has a high level of evidence (GRADE 1+/ strong agreement). Also, PDT is associated with a shorter operative time and a decreased incidence of stoma infection, inflammation and post-procedural major bleeding [Trouillet et al. 2014, Annals of Intensive Care]. (added on raws 321-325 of the manuscript)
  • All patients were treated with therapeutic dose anticoagulants, and 23 patients (21%) presented spontaneously resolved minor bleeding at the stoma site (of them, nine were treated in association with antiplatelet drugs). (added on raw 268-271) 
  • 342 (76%) patients went through OTI and of these, 133 patients (38.9%) underwent bedside PDT. (added on raw 92 and 190, and rectify the Figure 1.)
  • In the beginning of our study, we compared the difference in FiO2, PEEP and P/F ratio, between early and late group, but we found no significant data, and if we were to add it, the study would become confusing, inconvenient in the graphs and also from a clinical point of view.
  • Unfortunately, it is difficult to make a diagnosis of VAP in COVID-19 patients as the lung parenchyma in most cases has diffuse consolidations and can sometimes change over time regardless of superinfections. BAL positivity could be considered but even then we could not make a diagnosis of VAP without detailed radiological investigations. Reason why we used in the discussion, examples of previous studies on ICU VAPs in pre-COVID era.

  • There was no statistically significant difference in the mortality between early (23 patients, 37.7%) and late group (19 patients, 32.2%) (p=0.165). (raw 294-295)

    In the attached manuscript, we've highlighted the changes in yellow.

    Please feel free to contact me if you have any question.

    Kind regards,

    Nardi Tetaj

Reviewer 3 Report

I read the work “Outcomes and timing of bedside percutaneous tracheostomy of COVID-19 patients over a year in Intensive Care Unit” with pleasure. I absolutely agree that recommendations on both safety and timing protocols in tracheostomy in COVID-19 patients are still controversial issues. In this respect, this publication is very interesting. According to the "Type of Study", these results qualify as "Case series, low-quality cohort or case-control studies", which means that according to the Grades of Recommendation, they are only "C" level and 4th level according to the Level of Evidence. Undoubtedly, this topic requires further research on larger groups of patients. In my opinion, it is a very well-written research paper in all respects. However, it must be remembered that the work performed has a number of essential limitations, some of which the authors discussed in "Conclusions".

Author Response

Dear Reviewer, we thank you for your valuable suggestions. We took your advice and made changes on the manuscript. 

In the attached manuscript, we've highlighted the changes in yellow.

Please feel free to contact me if you have any question.

Kind regards,

Nardi Tetaj

Round 2

Reviewer 3 Report

I have no new comments

Author Response

Dear Reviewer,
Thank you again for your appreciated advice. We made a new modification of the data in relation to patient numbers as request from Reviewer 1, who suggested to include in the statistical analysis of the ICU LOS and total mechanical ventilation the 10 patients died within 7 days from the PDT, (which were excluded in the first version to simplify the study, so that the sample remains the same even in the evaluation of drug dosages on the seventh day of the PDT, i.e. 110 patients instead of 120). 

We also modified the conclusions and limits of the study as requested by you.
The revised manuscript is upload.
If needed, we are available to send the first version of the analysis in the supplement, i.e. the one with 110 patients where the 10 patients who died within 7 days from PDT were excluded.

Please do not hesitate to contact us if you have any further recommendation.
Waiting for Your kind reply please accept our best regards
Nardi Tetaj
